# Modeling-Based Decision Support System for Radical Prostatectomy Versus External Beam Radiotherapy for Prostate Cancer Incorporating an *In Silico* Clinical Trial and a Cost–Utility Study

**DOI:** 10.3390/cancers13112687

**Published:** 2021-05-29

**Authors:** Yvonka van Wijk, Bram Ramaekers, Ben G. L. Vanneste, Iva Halilaj, Cary Oberije, Avishek Chatterjee, Tom Marcelissen, Arthur Jochems, Henry C. Woodruff, Philippe Lambin

**Affiliations:** 1The D-Lab, Department of Precision Medicine, GROW—School for Oncology and Developmental Biology, Maastricht University, 6229 ER Maastricht, The Netherlands; i.halilaj@maastrichtuniversity.nl (I.H.); c.oberije@maastrichtuniversity.nl (C.O.); a.chatterjee@maastrichtuniversity.nl (A.C.); a.jochems@maastrichtuniversity.nl (A.J.); h.woodruff@maastrichtuniversity.nl (H.C.W.); philippe.lambin@maastrichtuniversity.nl (P.L.); 2Department of Clinical Epidemiology and Medical Technology Assessment (KEMTA), Maastricht University Medical Centre+, 6229 HX Maastricht, The Netherlands; bram.ramaekers@mumc.nl; 3Department of Radiation Oncology (MAASTRO), GROW—School for Oncology and Developmental Biology, Maastricht University Medical Center+, 6229 HX Maastricht, The Netherlands; ben.vanneste@maastro.nl; 4Department of Urology, Maastricht University Medical Centre+, 6229 HX Maastricht, The Netherlands; tmarcelissen@gmail.com

**Keywords:** prostate cancer, decision support system, in silico trial, cost-effectiveness, radical prostatectomy, external beam radiotherapy

## Abstract

**Simple Summary:**

Low–intermediate prostate cancer has a number of viable treatment options, such as radical prostatectomy and radiotherapy, with similar survival outcomes but different treatment-related side effects. The aim of this study is to facilitate patient-specific treatment selection by developing a decision support system (DSS) that incorporates predictive models for cancer-free survival and treatment-related side effects. We challenged this DSS by validating it against randomized clinical trials and assessing the benefit through a cost–utility analysis. We aim to expand upon the applications of this DSS by using it as the basis for an in silico clinical trial for an underrepresented patient group. This modeling study shows that DSS-based treatment decisions will result in a clinically relevant increase in the patients’ quality of life and can be used for in silico trials.

**Abstract:**

The aim of this study is to build a decision support system (DSS) to select radical prostatectomy (RP) or external beam radiotherapy (EBRT) for low- to intermediate-risk prostate cancer patients. We used an individual state-transition model based on predictive models for estimating tumor control and toxicity probabilities. We performed analyses on a synthetically generated dataset of 1000 patients with realistic clinical parameters, externally validated by comparison to randomized clinical trials, and set up an in silico clinical trial for elderly patients. We assessed the cost-effectiveness (CE) of the DSS for treatment selection by comparing it to randomized treatment allotment. Using the DSS, 47.8% of synthetic patients were selected for RP and 52.2% for EBRT. During validation, differences with the simulations of late toxicity and biochemical failure never exceeded 2%. The in silico trial showed that for elderly patients, toxicity has more influence on the decision than TCP, and the predicted QoL depends on the initial erectile function. The DSS is estimated to result in cost savings (EUR 323 (95% CI: EUR 213–433)) and more quality-adjusted life years (QALYs; 0.11 years, 95% CI: 0.00–0.22) than randomized treatment selection.

## 1. Introduction

Prostate cancer (PCa) is the second most commonly diagnosed cancer for men, accounting for 13.5% of new cancers diagnosed in 2018 in the Netherlands, of which 40% are low- to intermediate-risk localized PCa [1,2]. PCa is a topic of heightened research interest, with new biomarkers and treatment modalities being tested at a high rate [3]. The leading choices for managing clinically localized PCa are external beam radiotherapy (EBRT), radical prostatectomy (RP), brachytherapy, and active surveillance [4]. For low- and intermediate-risk PCa, active surveillance is often proposed (in ~70% and ~30% of cases, respectively), and of the active treatment options, RP and EBRT are recommended most often (~50% and ~45%), according to the Netherlands Cancer Registry. However, no consensus has been reached as to which is superior in terms of effectiveness and/or toxicity, both due to the varying spectrum of toxicities as well as the difference in incidence. The treatment decision is often based on doctor preference and, to a much lesser extent, on patient preferences and patient-specific characteristics or expected outcomes such as (biochemical recurrence-free) survival or toxicity [5]. Typically, cost-effectiveness is not taken into account in the treatment decision. Since no calculations are performed based on patient-specific outcomes, one could argue that this treatment selection method is random when considered from an outcome perspective.

Previous work has compared EBRT to RP in terms of long-term survival as well as different toxicities. Chen et al. performed a meta-analysis of the efficacy of EBRT versus RP [6]. They reported no statistical difference in cancer-specific survival for low- and intermediate-risk patients. Potosky et al. [7] and Donovan et al. [8] showed that urinary incontinence and reduced sexual function were more common for RP, but bowel toxicity was more likely after EBRT. Currently, there are no studies available that help assess the individual benefits of EBRT versus RP based on patient characteristics [9,10,11], even though it has been suggested that parameters such as age, BMI, tumor grade, and pretreatment prostate-specific antigen (PSA) levels do influence both recurrence-free survival and toxicity [12].

The importance of personalized medicine has become progressively evident, and treatment selection for PCa is no exception [13]. An important step towards personalized PCa treatment would be a clinical decision support system (DSS), introduced in 2013 [14], that aids in the decision between RP and EBRT. Additionally, when considering the limited resources available for cancer care, it is becoming increasingly important to consider the cost–benefit ratio when comparing treatments to guide the decision-making process [15]. The integration of a clinical DSS could aid in this, as it has already for proton therapy [16].

In addition, the possibility of very cost-effective in silico trials (individualized computer simulations used in the development of drugs, devices, or interventions) promise to improve clinical research through better design, more transparent and detailed information about possible results, and greater explanatory power in interpreting side effects, as well facilitating the exploration of interactions with the individuals’ biology and the long-term or rare effects.

Our hypothesis is that the DSS can accurately replicate the results from published studies and can be used for in silico trials. We also hypothesize that the use of a DSS for treatment selection results in better tumor control, less toxicity, increased patient quality of life (QoL), and improved cost–benefit ratio when compared with current clinical practice based on tumor boards or medical specialist opinion.

The aim of this study is to build such a DSS using predictive models for estimating tumor control and toxicity probabilities for both RP and EBRT for low- to intermediate-risk localized PCa patients and validate this by comparing it to published clinical trials. We also set up an in silico trial using this model-based approach to assess the outcome for elderly patients. Additionally, we compared the cost-effectiveness (CE) of applying this DSS to random treatment decisions as a proxy for current clinical practice (Figure 1). The DSS will be made available at www.ai4cancer.ai (accessed on 25 March 2021).

## 2. Materials and Methods

### 2.1. Decision Support System

#### 2.1.1. Markov Model

The target population consisted of overall tumor stage T1-T2 PCa patients who were eligible for active treatment (i.e., EBRT and RP). The DSS was developed by constructing an individual state-transition model to estimate the effects and associated costs of treatment with RP vs. EBRT for each patient. Based on patient-specific parameters (e.g., age) and treatment type (EBRT or RP), probabilities of developing long-term toxicities, including rectal bleeding, urinary incontinence, and impotence or a combination, are calculated. After treatment, patients have a risk of progressing to the recurrence state, which is dependent on patient-specific parameters (e.g., Gleason score), after which they can develop metastatic disease and subsequently progress to PCa-related death. Furthermore, from any health state, it is possible to die of causes unrelated to cancer (Figure 1). The DSS then provides a comparison of tumor control probability (TCP), probability of chronic erectile dysfunction (ED), chronic urinary incontinence (UI), and late rectal bleeding (RB), as well as a comparison of expected costs and quality-adjusted life years (QALYs). Detailed explanations of the transition probabilities are shown in Appendix A.

Several assumptions are made in this model, the four most relevant being: (i) All RP patients start with ED and UI (at the first cycle); (ii) when a patient has developed biochemical failure, the utility value is described by a single number rather than separate disutility for ED, UI, and RB; (iii) we assume that after progression, the patient cannot return to the cancer-free health state; (iv) the transition probability from recurrence to metastatic disease is the same for all patients for both treatments. Moreover, a cycle time of one month and a time horizon of 20 years were used; these were chosen since the survival of low–intermediate prostate cancer is high, while increasing the time horizon beyond 20 years would unnecessarily increase model uncertainty.

In order to quantify the relative importance of various health outcomes using a common measurement unit, utility is used as a metric to assign weights to health states on a scale ranging from 0 (for dead) to 1.0 (for perfect health). Health-state-specific utilities, also incorporating treatment-related toxicities (and all possible combinations), were retrieved from Stewart et al. [17]. QALYs are obtained by multiplying these utility values by the time spent in the corresponding health state.

In order to account for the baseline utility of men living in the Netherlands, we used a published model that had calculated the age-dependent health-related quality of life (HRQoL) for different countries and applied it as a multiplicative factor to health-state-specific utilities [18].

The costs were calculated from the perspective of the healthcare system, so other societal costs, such as productivity losses and patient and family costs, were excluded. For detailed descriptions of the utility and cost data, see Appendix A.

#### 2.1.2. Predictive Models

The transition probabilities were estimated per individual in order to make this DSS patient-specific and ready for precision medicine applications. The individual probabilities of progression after treatment and the risk of developing toxicities were calculated using a selection of regression models or nomograms from the published literature (Table 1), adherent to the TRIPOD statement [19]. For nomograms, the coefficients or intercepts were derived (if not reported) by reading the nomogram and using interpolation and fitting.

### 2.2. Validation

#### 2.2.1. Synthetic Dataset

Since no complete patient cohort for which all required input parameters were known was identified, we generated a synthetic patient dataset. A patient is described by a set of parameters, which we randomly assigned by drawing values from a distribution. We chose the distributions based on the patient cohorts on which the models had been developed (Appendix A). We assumed that the clinical parameters were independent of one another and generated a synthetic patient dataset including 1000 patients so that the mean and associated error of the generated clinical parameters matched that of the original datasets.

#### 2.2.2. Validation of NTCP and TCP Models

In order to validate the combination of our NTCP and TCP models, we compared the simulation results to the published results. The reliability of the model and the measure in which the synthetic dataset reflects real patient datasets were assessed by generating the synthetic patients so that they matched the reported clinical parameters from actual trials, such as age, Gleason score, PSA values, and T-stage. Nonreported clinical parameters were kept the same as the original synthetic dataset. Dosimetric parameters were scaled according to the 2 Gy equivalent dose. The relevant outcomes, such as biochemical free survival or toxicity, were then compared between the simulation and the clinical trial. For the EBRT α/β ratio (a measure of the fractionation sensitivity of the tissue) of PCa, we used a value of 1.5 Gy [26].

In order to assess the credibility of the predicted biochemical free survival, the model was compared to the results of a randomized clinical trial, reported by Hamdy (2016), that compared EBRT to RP [27]; to assess the effect of hypofractionation, we compared biochemical free survival to the CHHiP trial [28]. In order to validate the model’s predictions of toxicity, we compared the model results to Donovan et al. (2016), who published patient-reported outcomes after EBRT or RT [8].

#### 2.2.3. In Silico Trial

A DSS such as this, in combination with the synthetic patient dataset, could function as an in silico clinical trial, a precursor to actual clinical trials, in order to improve study design or explanatory power. We demonstrated this by generating a patient dataset with patients aged 75–80 to test the outcomes of different treatments for elderly patients, an often-underrepresented group in clinical trials, but a group that might still be eligible for both treatments. We adjusted pretreatment erectile function to an average of 15% to reduce the impact of ED on the outcome.

### 2.3. Cost-Effectiveness Analysis

In order to determine the potential benefit of a DSS for PCa, we performed CE analyses that compared two different treatment allotment strategies. In the first, each patient in the synthetic dataset was allotted the treatment for which the DSS calculated the highest number of QALYs. For the second, each patient was allotted a randomized treatment as a proxy for current clinical practice.

To determine which treatment allotment strategy was cost-effective, the incremental net monetary benefit (iNMB) was calculated. This is described by the following equation:(1)$$iNMB=\left(QALY_{DSS}-QALY_{random}\right)\cdot WTP-(Costs_{DSS}-Costs_{random})$$
where $Costs_{DSS}$ and $Costs_{random}$ are the mean costs per patient if treatment decisions are based on a *DSS* or randomized treatment allotment strategy, respectively, and $QALY_{DSS}$ and $QALY_{random}$ are the mean number of *QALYs* per patient. WTP is the willingness to pay per *QALY*. If the *iNMB* is positive, the *DSS* is considered cost-effective. In the Netherlands, a *WTP* threshold of EUR 80,000 is agreed upon for high burden diseases [29]. We applied discount rates of 1.5% for the *QALYs*, and 4% for the costs, which is standard practice within the Netherlands [30].

## 3. Results

### 3.1. Synthetic Data Characteristics

When applying the DSS on a synthetically generated patient dataset of 1000 patients with clinical parameters similar to those on which the predictive models were built, 47.8% of the patients had a higher number of QALYs for RP and 52.2% for EBRT. The patients for whom EBRT was chosen had a higher mean age (64 versus 59 years), higher mean prostate-specific antigen (PSA) values (8.3 versus 6.8 ng/mL), and a higher percentage of T2 stage (49% versus 24%), as shown in Table 2 and Appendix A.

### 3.2. Validation

We compared the outcome reported in three separate papers against the same outcome simulated by our DSS on a synthetic patient dataset; the results are reported in Table 3. More details on the parameter values used are shown in Appendix A, and results are shown in Figure 2. The progression-free survival reported by Hamdy (2016) was very similar to our simulation results, as was the Kaplan–Meier curves obtained from the GUROC (Genitourinary Radiation Oncologists of Canada) [31] Prostate Cancer Risk Stratification (ProCaRS) database [32], and the National Prostate Cancer Register of Sweden (NPCR) [21,33]. Dearnaley (2016) published the results of the CHHiP trial and compared the results of different fractionation plans [28]. When comparing these results to what was simulated by our DSS, the simulation was close to the published results, but the difference increased with stronger hypofractionation. Donovan (2016) was used for toxicity comparisons and showed that the DSS simulated late toxicity best but was less accurate for acute toxicity. The relative differences between the simulations and the studies for the different treatment modalities were similar, and the conclusions coincided. The most notable discrepancy is for acute ED for EBRT, which was underestimated by the simulations because we assumed no acute ED after EBRT treatment.

### 3.3. In Silico Trial

We performed an in silico trial on the synthetic dataset by increasing the age of all patients to between 75 and 80 but leaving all other clinical parameters as they were. We found that TCP for RP was marginally higher than for EBRT (HR: 1.007), and the risk of chronic UI was much higher for RP (HR: 10) as well as ED (HR: 1.59); consequently, EBRT did result in more QALYs and lower costs than RP. The DSS selected EBRT for 96% of the patients. We repeated this analysis while assuming that 85% of the patients had ED before the start of treatment. This resulted in a smaller difference in ED, thus decreasing the difference in QALYs between RP and EBRT (Table 4) and resulting in the DSS selecting EBRT for 72% of the patients.

### 3.4. Cost-Effectiveness Analysis

When comparing the DSS treatment selection to a randomized treatment allotment strategy, the DSS resulted in an average cost saving (discounted) of EUR 317 per patient and an increase in the number of discounted QALYs of 0.11 years. With an incidence rate of 12,500 patients a year in the Netherlands, with 40% being low- to intermediate-risk PCa patients and 50% receiving active treatment, this DSS could affect ~2400 patients a year. Since the costs decreased while the QALYs increased, the DSS was the dominant strategy in this “base case” scenario, with an iNMB (Equation (1)) of EUR 8798 per patient. The TCP is higher for the DSS than for randomized treatment allotment, and the probability of all toxicities decreases with the use of this DSS (Table 5).

A probabilistic sensitivity analysis, performed with 5000 Monte-Carlo simulations, resulted in a cohort with a mean increase in discounted QALYs of 0.11 (95% CI: 0.00–0.22) and a mean decrease in costs of EUR 323 (95% CI: EUR 213–433) (Figure 3) for DSS compared with randomized treatment allocation. For 98.6% of the simulations, using a DSS was cost-effective when compared to a randomized strategy (dominant for 98.2% of the simulations). For a detailed cost-effectiveness analysis of the model, see Appendix A.

## 4. Discussion

In this study, we developed a clinical DSS for the treatment of PCa patients with either EBRT or RP and tested this on a synthetic patient dataset. We validated the DSS against published clinical studies and set up an in silico trial for patients between 75 and 80, eligible for both RP and EBRT. We also assessed the CE of a treatment allotment strategy based on the DSS compared to a randomized treatment allotment strategy. Our first hypothesis was that we could accurately replicate results from published studies, which we aimed to confirm by generating synthetic datasets with clinical parameters similar to published trials. The DSS largely replicated the published results accurately. The relative differences between the treatment modalities and fractionation plans were replicated by the model, and the conclusions of the DSS and the studies agreed. We also performed an in silico trial, exclusively including elderly patients first without, then with, prior ED, using the DSS and found that for the first group, EBRT was preferred, and, for the second, RP performed better in terms of QALYs. Additionally, we hypothesized that a treatment selection strategy based on the DSS would improve tumor control, reduce toxicity, and improve CE as opposed to randomized treatment selection. Our CE analyses suggest that not only do the costs of treatment decrease with the application of a DSS, but the number of QALYs also increases, making the integration of a DSS dominantly cost-effective compared to current clinical practice. The expected cost savings within the Netherlands when using a DSS could be as high as EUR 3.8 million over five years, assuming 2400 patients are affected every year. Additionally, the number of patients with recurrence after treatment could be reduced slightly by 2%.

These results imply that when deciding between RP and EBRT for a given patient, making the right choice can improve overall QoL and that this decision should not be random. The DSS offers the possibility of combining a large number of clinical parameters to predict NTCP and TCP and quantify these risks into a single metric for different treatment options. This has the potential to improve the decision-making process, along with other factors such as incorporating patient preferences. The development of a DSS fits well into the current trend that strives for personalized medicine, and the results presented in this study confirm the added benefit of such a tool [34]. The application of the DSS for in silico trials has great potential benefits; it not only improves the design of clinical trials through precursory simulations but also has the benefit of being able to apply different treatments to the same “patient”, which allows for a more objective comparison. Another advantage of the DSS is that it is detailed and can be further extended with other disease management options such as brachytherapy or active surveillance. The framework of the DSS is also flexible, making it easy to replace individual models or update them as new clinical trial results are published. It can also be used as the basis for an individualized patient decision aid (iPDA).

This study has several limitations. The first one is that this is a model-based study, using models that were trained and validated on different cohorts. The models were selected based on how recently they were published, the number of patients included, and whether or not they used clinical parameters and a TRIPOD level. We also attempted to make sure we only selected models trained on patients with similar treatment modalities and similar clinical parameters; however, not all clinical parameters were reported. In addition, the correlation between clinical parameters was not reported, and when generating the synthetic dataset, no correlations were assumed. It should also be noted that the clinical parameters found in the clinical trials used as a basis for the synthetic dataset may not be representative of the whole patient population. Different outcomes of the models were validated in different studies, so the DSS as a whole has not been validated on a single patient population. However, the acquisition of a dataset where not only all the clinical parameters but also long-term follow-up data for TCP and toxicity for both treatment arms are reported might not be feasible. Moreover, the NTCP models used doctor-reported outcomes as endpoints, while the validation was done on patient-reported outcomes. However, the promising benchmark results indicate that the models have significant value in simulating reality. These results were generated using an α/β ratio of 1.5; should this value be proven inaccurate, these results will have to be reassessed.

In the CE analysis, we used randomized treatment allotment as a proxy for clinical practice; however, we were not able to compare the performance of the CE model to actual clinical practice. This is because there is no hard baseline and actual clinical practice can vary strongly between countries and even hospitals. However, in Appendix A, we do compare the performance of the CE model to simple clinical parameters, which, in clinical practice, could help determine treatment choice. In its current state, the DSS does not take into account patient preferences but uses average utility values obtained from a population. However, the risks of different types of toxicities are what often drive treatment decision-making, and patient preferences should be taken into account. Future versions of the DSS should allow patients’ input to affect the utility values of different toxicities and be integrated into iPDAs to allow for shared decision-making [35,36]. This is especially true for the 15% of patients for whom the expected difference in QALYs is very low (<0.05 years).

A limitation of the reported CE analysis is that both the DSS and the CE model were based on the same state-transition model (committing a potential petitio principii fallacy). The differences between the two were that the DSS was based on the undiscounted deterministic run, while the CE model used discount rates for both QALYs and costs and was based on probabilistic simulations. The costs used in this study were based on the health costs in the Netherlands. One of the assumptions in the model was that if the patient had biochemical failure after primary treatment, no secondary treatment would take place, though, often, this is not the case. The health-state recurrence now has the same costs as a regular follow-up, but with secondary treatment, these costs would be higher. However, when applying the model with higher costs for recurrence, matching them with the costs for metastatic disease, the cost-effectiveness was not impacted, so this assumption has no direct impact on our conclusions. For the application of the tool in other countries, the costs will be most complex to adapt. However, utility values are simpler to adjust to different countries, so it might be possible only to transfer the section of the tool that calculates QALYs, which already provides valuable information that can support decision making. However, it should be aspired to include the monetary calculations, as these play an increasingly large role in healthcare decision making.

Short-term future work includes merging this DSS with previously developed DSSs, including one where the effectiveness of an implantable rectum spacer is accessed on the patient level [37,38] as well as a DSS that compares photon to proton therapy. We also plan to expand the NTCP models by including single nucleotide polymorphisms and tumor somatic mutations to incorporate genetic information into the decision-making process [39].

## 5. Conclusions

This study lays the groundwork for a detailed, personalized treatment DSS that aids in the choice between EBRT and RP for low- to intermediate-risk PCa patients. This DSS could be used for in silico clinical trials when applied to a synthetic dataset, which would be a valuable precursor to clinical trials. The results suggest that the full development and clinical application of this DSS would improve the quality of patient care as well as cost-effectiveness and would be an important step towards personalized and participative treatment decisions.

## Figures and Tables

**Figure 1 cancers-13-02687-f001:**
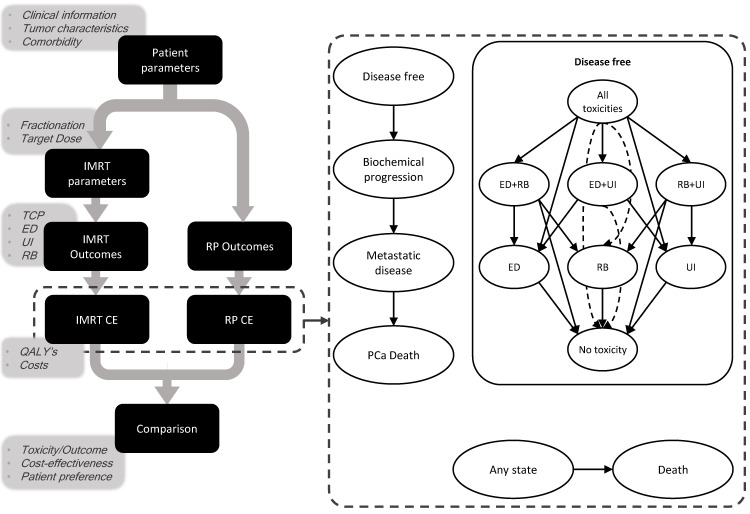
Setup of the clinical decision support system and a summary of the Markov model. Ovals represent different health states; arrows represent transitions between health states. Dashed arrow lines are for intelligibility purposes. Patients start in the disease-free state, with either all toxicities, ED and UI, or UI only, and as time passes, they can recover from toxicity or progress into the biochemical progression state. Death unrelated to cancer can occur from any health state; cancer-related death only from the metastatic disease state. IMRT: intensity-modulated radiotherapy, RP: radical prostatectomy, CE: cost-effectiveness, QALY: quality-adjusted life year, TCP: tumor control probability, ED: erectile dysfunction, UI: urinary incontinence, RB: rectal bleeding, PCa: prostate cancer.

**Figure 2 cancers-13-02687-f002:**
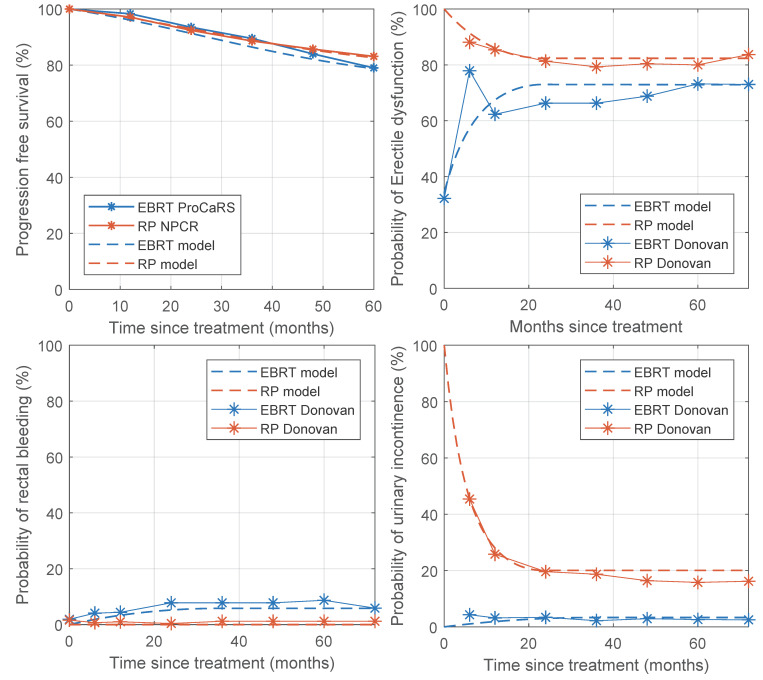
Progression-free survival and probability of toxicity of the model compared to published data. Progression-free survival data were obtained from the GUROC (Genitourinary Radiation Oncologists of Canada) Prostate Cancer Risk Stratification (ProCaRS) database and the National Prostate Cancer Register of Sweden (NPCR). Data on toxicity was obtained from Donovan et al. (2016).

**Figure 3 cancers-13-02687-f003:**
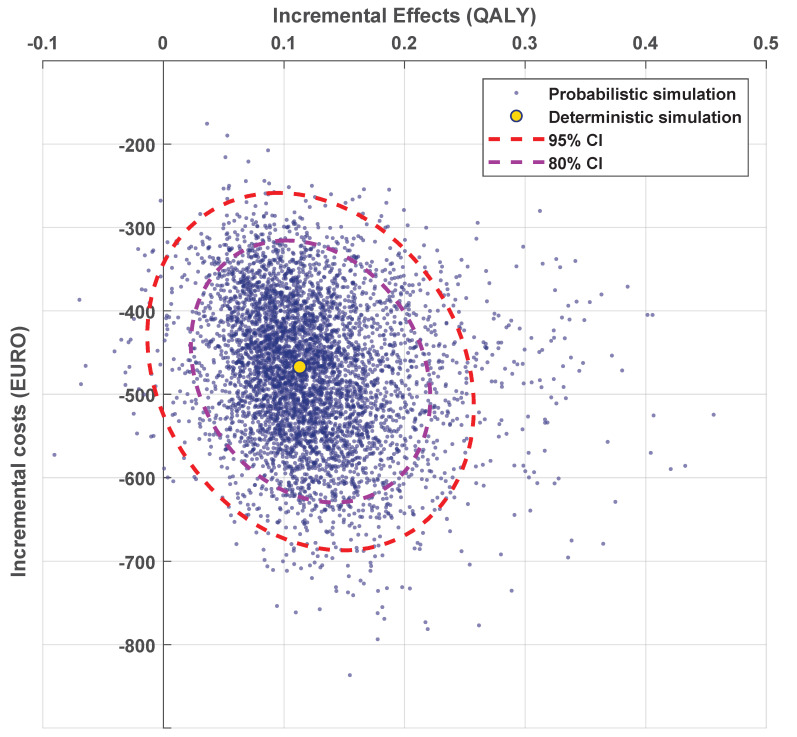
The cost-effectiveness plane when comparing the DSS to the randomized strategy. Each blue point represents the incremental costs and quality-adjusted life years (QALYs) for DSS vs. randomized treatment allocation of a single simulation over all 1000 patients. The yellow dot shows the deterministic simulation, and the dashed lines show the confidence intervals of the simulations.

**Table 1 cancers-13-02687-t001:** An overview of the literature models used for the state transition probabilities. Rectal bleeding does not typically occur after RP, so the transition was set to zero for this treatment type.

Model	TRIPOD	Treatment (n)	Parameters	Outcome	Performance
Warner et al. 2015 * [20]	1a + 4	EBRT (822 + 967)	ADT [months] PSA [ng/mL] Gleason score BED [Gy]	5-year BFFS	R^2^ = 0.868
Bjartell et al. 2016 [21]	3	RP (3452 + 1762)	T-stage [T1/T2] PSA [ng/mL] Primary Gleason grade Secondary Gleason grade Positive biopsy cores [n] Negative biopsy cores [n]	5-year RFS **	C-index = 0.68
Schaake et al. 2018 [22]	1b	EBRT (243)	Mean Trigone dose [Gy]	Late UI after 3 years	AUC = 0.66
Matsushita et al. 2015 * [23]	2a	RP (2849)	Age [years] BMI [kg/m^2^] ASA score [I/II/III/IV] Urethral length [mm]	Recovery from UI after 1 year	AUC = 0.71
Alemozaffar et al. 2011 [24]	3	Both (524 + 241)	Age [years] Nerve-sparing [y/n] PSA [ng/mL] ADT [y/n]	Erection recovery after 2 years	AUC = 0.77 for RP, AUC = 0.87 for EBRT
Liu et al. 2010 [25]	2a	EBRT (161)	DVH V75	Late RB after 3 years	AUC = 0.62

TRIPOD: Transparent reporting of a multivariable prediction model for individual prognosis or diagnosis; EBRT: external beam radiotherapy; RP: radical prostatectomy; n: number; ADT: androgen deprivation therapy; PSA: prostate-specific antigen; BED: biologically effective dose; BFFS: biochemical failure-free survival; R2: coefficient of determination; Gy: gray; BMI: body mass index; C-index: concordance statistic; RFS: regression free survival; UI: urinary incontinence; ASA: American Society of Anesthesiologists; V75: volume receiving at least 75 Gy; RB: rectal bleeding. * Due to incomplete information, some model coefficients had to be derived from the nomogram. ** Recurrence was defined as biochemical failure, initiation of secondary therapy, distant metastases, or prostate-cancer-related death.

**Table 2 cancers-13-02687-t002:** Patient characteristics of the synthetic patient dataset.

Parameter Name	EBRT Mean (SD)	RP Mean (SD)	*p*
Age (years)	63.8 (10.7)	58.8 (9.1)	<<0.001
PSA (ng/mL)	8.3 (3.5)	6.8 (3.5)	<<0.001
T-stage 2 (%)	48.7	24.4	<<0.001
P. Gleason grade 4 (%)	27.1	3.3	<<0.001
S. Gleason grade 4 (%)	14.7	20.9	0.01
ADT given (%)	0	80.0	0.72

PSA: prostate-specific antigen; T-stage: tumor stage; P.: primary; S.: secondary; ADT: androgen deprivation therapy.

**Table 3 cancers-13-02687-t003:** Validation on published clinical trial results.

Study	Treatment	Outcome	Clinical Trial (%)	Simulation (%)	Difference (%)
Hamdy 2016 [27]	EBRT [545]	5 year BDFS	88.5	89.0	0.5
	RP [553]	5 year BDFS	88.1	87.0	1.1
CHHiP [28]	EBRT 74 Gy [1065]	5 year BDFS	88.3	88.4	0.1
	EBRT 60 Gy [1074]	5 year BDFS	90.6	89.9	0.7
	EBRT 57 Gy [1077]	5 year BDFS	85.9	87.9	2.0
Donovan 2016 [8]	EBRT [545]	6 month UI	5	3.3	1.7
		6 year UI	3.5	3.3	0.2
		6 month ED	77.8	57.3	20.5
		6 year ED	73.0	72.9	0.1
		6 month RB	3.8	1.9	1.9
		6 year RB	5.9	5.8	0.1
	RP [553]	6 month UI	46	45.5	0.5
		6 year UI	17	20.1	3.1
		6 month ED	88	91.3	3.3
		6 year ED	83.5	82.4	1.1

N: number of patients; EBRT: external beam radiotherapy; RP: radical prostatectomy; BDFS: biological disease-free survival; UI: urinary incontinence; ED: erectile dysfunction; RB: rectal bleeding; Gy: gray; GS: Gleason score.

**Table 4 cancers-13-02687-t004:** In silico trial results.

Outcome Type	RP Mean (SD)	EBRT Mean (SD)	DSS Mean (SD)	RP PTED Mean (SD)	EBRT PTED Mean (SD)	DSS PTED Mean (SD)
5 year TCP [%]	84 (7)	84 (4)	84 (4)	84 (7)	84 (4)	85 (5)
2 year ED [%]	93 (6)	61 (19)	61 (20)	100 (0)	98 (2)	98 (2)
1 year UI [%]	70 (16)	7 (4)	8 (10)	70 (16)	7 (4)	22 (26)
3 year RB [%]	0 (0)	5 (4)	5 (4)	0 (0)	5 (4)	3 (4)
QALY	4.28 (0.87)	4.46 (0.86)	4.46 (0.86)	4.25 (0.85)	4.30 (0.83)	4.32 (0.84)
Costs (1000 €)	15.2 (1.1)	12.6 (1.0)	12.7 (1.1)	15.2 (1.1)	12.7 (1.0)	13.3 (1.5)

RP: radical prostatectomy; EBRT: external beam radiotherapy; DSS: decision support system; PTED: 85% pretreatment erectile dysfunction; SD: 1 standard deviation; TCP: tumor control probability; ED: erectile dysfunction; UI: urinary incontinence; RB: rectal bleeding; QALYs: quality-adjusted life years.

**Table 5 cancers-13-02687-t005:** Probability of different outcomes.

Outcome Type	RP Mean (SD)	EBRT Mean (SD)	*p*	Random Mean (SD)	DSS Mean (SD)	*p*
5 year TCP [%]	83 (8)	84 (4)	0.003	83 (7)	85 (5)	<<0.001
2 year Erectile Dysfunction [%]	83 (13)	59 (2)	<<0.001	71 (14)	67 (20)	<<0.001
1 year Urinary Incontinence [%]	27 (18)	7 (4)	<<0.001	16 (9)	13 (15)	<<0.001
3 year Rectal Bleeding [%]	0 (0)	5 (4)	<<0.001	2 (2)	1 (3)	0.61

RP: radical prostatectomy; EBRT: external beam radiotherapy; DSS: decision support system; SD: standard deviation; TCP: tumor control probability.

## Data Availability

Not Applicable.

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
