# Peer review of "Modeling-Based Decision Support System for Radical Prostatectomy Versus External Beam Radiotherapy for Prostate Cancer Incorporating an In Silico Clinical Trial and a Cost–Utility Study"

_cancers, 2021, doi:10.3390/cancers13112687_

Round 1

Reviewer 1 Report

I appreciate this paper, which is well written, for the objective and the methodological approach. I have a few minor questions and comments.

I. None of the figures and tables were referenced to in the manuscript.

2. Several references are missing. In particular, references to GURCOC, ProCaRS and NPCR data presented in Figure 2.

3. You choose a value of alfa/beta of 1.5 Gy. What will the outcome be in  Table 3 with a more conservative value of 3.0 Gy?

4. Concerning the patient characteristics for the synthetic patient data set you assume that close to 80 % of the patients receive ADT both with EBRT and RP. I think these figures are unrealistic for low-intermediate risk tumors. In general, ADT is not used with prostatectomy. How do you motivate this assumption? Related to this, I also question the information in the table in APP IV concerning the length of ADT, which is assumed to be 19.0 months for EBRT and 4.1 months for RP. Motivation?

5. Validation. Lines 245 to 247. This statement is not in agreement with the figures presented in Table 3 concerning the CHHiP trial and the simulation. The difference to hypofractionation is 0.7 % in the Table, but in the text you say 7 %. Explanation?

6. In Silico trial results, Table 4, The headings RP mean low ED, EBRT mean low ED and DSS mean low ED are confusing and unclear.

7. Cost effectiveness. On the line 269 you say that the average cost saving is 317 euro per patient. I do not understand how do you derive the value 8798 euro per patient on line 274. Please clarify.

Reviewer 2 Report

The authors tried to evaluate quality of their decision support system for patients with low-intermediate risk prostate cancer, when they select RP or EBRT as an appropriate treatment.

  • The authors made some assumptions to build up DSS. The reviewer cannot understand the description “when a patient has developed biochemical failure, it is assumed that toxicity dose not affect the health-related quality of life (QOL) or the costs anymore”. When the patients had biochemical failure, they may receive salvage radiotherapy or hormonal therapy when they needed. At that time QOL may be deteriorated according to the treatment they receive. Please explain the issue for clarification.
  • In silico trial, the authors generated a patient dataset with patients aged 75-90. However, those patients may not receive radical prostatectomy in most cases.  The authors should explain the issue.
  • The authors presented the limitations of this study. However, the reviewer cannot agree with the description that “for the application of the tool in other countries, only the costs would need to be adjusted”. Sexual attitudes and QOL among races who received RP or EBRT may be different (PMID 14742098).  The authors should present those limitations.

Round 2

Reviewer 2 Report

  1. The reviewer cannot still agree with the idea that patients aged 75-90 are eligible to both EBRT and RP. If the patients are aged 75-80 RP may be acceptable and if the patients are aged 75-85 EBRT may be acceptable. The authors should re-evaluate according to the reviewer’s comment correctly.
  2. The reviewer cannot still agree with the explanation of the authors about Q3. Do you think that the description “the costs do not need to impact the decision making, (p11 L 372-373)” is really correct?

Please check the description below:

  • p3 L127-128: the description “when a patient has developed bio-127 chemical failure” is duplicated.
